# Design Analysis and Actuation Performance of a Push-Pull Dielectric Elastomer Actuator

**DOI:** 10.3390/polym15041037

**Published:** 2023-02-19

**Authors:** Wenjie Sun, Bin Zhao, Fei Zhang

**Affiliations:** 1School of Mechanical and Precision Instrument Engineering, Xi’an University of Technology, Xi’an 710048, China; 2State Key Laboratory for Strength and Vibration of Mechanical Structures and School of Aerospace, Xi’an Jiaotong University, Xi’an 710049, China; 3CAS Center for Excellence in Nanoscience, Beijing Key Laboratory of Micro-nano Energy and Sensor, Beijing Institute of Nanoenergy and Nanosystems, Chinese Academy of Sciences, Beijing 101400, China

**Keywords:** dielectric elastomer, push-pull DEA, key design parameters, dynamic response, natural frequency

## Abstract

Dielectric elastomer actuation has been extensively investigated and applied to bionic robotics and intelligent actuators due to its status as an excellent actuation technique. As a conical dielectric elastomer actuator (DEA) structure extension, push-pull DEA has been explored in controlled acoustics, microfluidics, and multi-stable actuation due to its simple fabrication and outstanding performance. In this paper, a theoretical model is developed to describe the electromechanical behavior of push-pull DEA based on the force balance of the mass block in an actuator. The accuracy of the proposed model is experimentally validated by employing the mass block in the construction of the actuator as the object of study. The actuation displacement of the actuator is used as the evaluation indication to investigate the effect of key design parameters on the actuation performance of the actuator, its failure mode, and critical failure voltage. A dynamic actuator model is proposed and used with experimental data to explain the dynamic response of the actuator, its natural frequency, and the effect of variables. This work provides a strong theoretical background for dielectric elastomer actuators, as well as practical design and implementation experience.

## 1. Introduction

As an emerging type of actuation, soft actuators have been widely explored and applied to intelligent actuators [1,2,3], deep-sea exploration [4,5], bionic robotics [6,7,8], tunable optical devices [9,10], and morphological control [11,12] due to their outstanding features such as softness and flexibility, strong environmental adaptability, and excellent biocompatibility. Among the various soft actuation principles [13], dielectric elastomers have attracted great attention due to their large actuation deformation, fast response, and high energy density [14]. Existing research has shown that dielectric elastomers can produce area strains of up to 380% [15], and even more than 2000% under voltage [16]. However, the particular actuation performance of dielectric elastomer actuators depends on the actuator structure. Currently, the most common structures for dielectric elastomer actuators are tubular [17], spherical [18], stacked [19], conical [20], and minimum-energy structures [21]. Among them is the conical dielectric elastomer actuator (conical DEA), which employs spring components or mass blocks to develop a flat sheet of elastomer film into a conical one in three dimensions. The actuator produces out-of-plane deformation along the axial direction when a voltage is applied to the conical dielectric elastomer [22,23]. This actuator structure is preferred for dielectric elastomers to achieve unidirectional actuation owing to its high output force, compact system, convenient preparation, and robust scalability.

When a dielectric elastomer film is used instead of the required spring element or mass block in a conical DEA, a novel actuation structure is developed that is often called a push-pull dielectric elastomer actuation (push-pull DEA) [24]. As an evolved version of the conical DEA, this design has found applications in various fields, including tactile display [25], fluid control [26], and multi-stable actuation [27]. Similarly to the push-pull DEA, the double conical actuation design integrates two conical DEA back-to-back via rigid support or magnets [28,29]. Bionic vehicles [30], binary actuation [31], and pipeline robots [32] have been implemented in a double conical actuator. Theoretical and experimental studies have been conducted to evaluate the actuation performance of conical DEAs, as reported in references [22,23,33,34,35]. For instance, an electromechanical coupling model, which was developed based on thermodynamics and continua mechanics for conical DEAs, was utilized to examine the non-uniform deformation and potential failure modes of the actuator [22,23]. The multi-mode dynamic response of conical DEAs under varying excitation frequencies was analyzed through experiments [33]. Additionally, an electromechanical-magnetic coupling model was established to study the bistable performance of conical DEAs under magnetic fields [34]. The actuation performance of conical DEAs was also analyzed using the finite element method under different excitation voltages [35]. However, research into the performance of push-pull DEAs remains limited both theoretically and experimentally. Therefore, in this paper, the push-pull DEA’s design analysis and actuation performance, along with its influencing factors, are investigated and analyzed from theoretical and experimental perspectives. This work provides a reliable theoretical background and practical experience for the future widespread use of conical DEA.

## 2. Design and Theoretical Model of the Push-Pull DEA

A circular dielectric elastomer film with an initial thickness of H_0_ is pre-stretched in an equal biaxial direction and then attached to a circular rigid frame with an inner radius of B. The purpose of applying a biaxial pre-stretch is to increase the electric field strength under a constant voltage by reducing the thickness of the dielectric elastomer film, thus enhancing the actuation performance of the DEA. The mass block with radius A is bonded to the film’s center to create a completed actuation part. Two actuation components of the same size are assembled and bonded at the mass block utilizing a rigid support with a length of L_0_. The compliant electrodes are painted on both sides of the film of one actuation component, which is defined as the active actuation part. On the other hand, the passive actuation part describes the other actuation component. The equal biaxial pre-stretching of the films of the active and passive parts are $\lambda_{a\left(p\right)}$ and $\lambda_{p\left(p\right)}$, respectively. The particular push-pull DEA preparation process used in this study, with experimental photographs, is shown in Figure 1.

We define the condition in which the mass blocks are not bonded as the pre-stretched state when the active and passive part film thicknesses are $H_{0}{\left(\lambda_{a\left(p\right)}\right)}^{-2}$ and $H_{0}{\left(\lambda_{p\left(p\right)}\right)}^{-2}$, respectively, as shown in Figure 2a. The reference state is established when the mass block of the active and passive parts is bonded together. The elastomeric films on both sides display a conical surface in space, and the displacement of the mass block M along the actuation direction is D_0_. The actuator produces the actuation deformation when a voltage Φ is applied to the active part. Parameter D is the new position of the mass block and $\lambda_{a\left(r\right)}$ and $\lambda_{a\left(c\right)}$ represent the current radial and hoop deformations of the active film, respectively. The radial and circular deformations of the passive film are correspondingly described by $\lambda_{p\left(r\right)}$ and $\lambda_{p\left(c\right)}$. This state is called the actuation state of the actuator. The force analysis diagram of push-pull DEA is depicted in Figure 2d. Since the mass of the elastomer film on both sides and the gravity effect are neglected, the analysis of the displacement of the mass block is employed to describe the electromechanical performance of the actuator under voltage excitation.

The thickness of the dielectric elastomer film in the conical DEA displays a nonlinear distribution in space under electromechanical coupling loading [22,23], and the actual shape of the dielectric elastomer film in the conical DEA is similar to a hyperbolic surface, formed by the solid gray line rotating 360 degrees around the actuation direction in Figure 2d. However, for the convenience of analysis, it is commonly simplified as a cone surface [30,34,36], which is formed by the red dotted line rotating 360 degrees around the actuation direction. Assuming that the forces exerted on the mass block by the active part and the passive part are $F_{a}$ and $F_{p}$, respectively, and the resistance of the mass block during the actuation process is represented by $\zeta \dot{D}$, where ζ represents the linear damping coefficient, the equilibrium control equation of the mass block at this moment is:(1)$$F_{p}sin\theta_{p}-F_{a}sin\theta_{a}-\zeta \dot{D}-M\ddot{D}=0$$
where $\theta_{a}$ and $\theta_{p}$ represent the inclination angle of the force exerted on the mass block by the active and passive parts, respectively, M is the total mass of the mass block, and D is its displacement.

The force $F_{a}$ applied on the mass block, the radial stress $\sigma_{a\left(r\right)}$ in the active part, the force $F_{p}$ applied on the mass block, and the radial stress $\sigma_{p\left(r\right)}$ in the passive part are related as follows [22]:(2)$$F_{a}=2\pi Ah_{a}\sigma_{a\left(r\right)}\;F_{p}=2\pi Ah_{p}\sigma_{p\left(r\right)}$$
where $h_{a}$ and $h_{p}$ denote the thickness of the active and passive part films in the actuation state, respectively. Assuming that the dielectric elastomer is an incompressible material, $h_{a}=H_{0}\lambda_{a\left(r\right)}^{-1}\lambda_{a\left(c\right)}^{-1}$, $h_{p}=H_{0}\lambda_{p\left(r\right)}^{-1}\lambda_{p\left(c\right)}^{-1}$. Based on the theory of dielectric elastomer [37,38], the radial stresses in the active and passive part films can be expressed as follows (see Appendix A):(3)$$\sigma_{a\left(r\right)}=\frac{\partial W_{a}\left(\lambda_{a\left(r\right)},\lambda_{a\left(c\right)}\right)}{\partial \lambda_{a\left(r\right)}}\lambda_{a\left(r\right)}-\frac{\epsilon \Phi^{2}}{H_{0}^{2}}\lambda_{a\left(r\right)}^{2}\lambda_{a\left(c\right)}^{2}\;\sigma_{p\left(r\right)}=\frac{\partial W_{p}\left(\lambda_{p\left(r\right)},\lambda_{p\left(c\right)}\right)}{\partial \lambda_{p\left(r\right)}}\lambda_{p\left(r\right)}$$
where ϵ is the dielectric constant of the dielectric elastomer, Φ represents the applied voltage to the active part, and $W_{a}$ and $W_{p}$ represent the strain energy density functions of the active and passive elastomer films, respectively.

Among the many functions that characterize the strain energy density of dielectric elastomers, the Gent free energy model is widely used for its ability to characterize the limit stretching properties of polymeric materials [39]. Detailed expressions of the Gent free energy model are shown below:(4)$$W\left(\lambda_{1},\lambda_{2}\right)=-\frac{\mu J_{lim}}{2}ln\left(1-\frac{\lambda_{1}^{2}+\lambda_{2}^{2}+\lambda_{1}^{-2}\lambda_{2}^{-2}-3}{J_{lim}}\right)$$
where μ and $J_{lim}$ denote the dielectric elastomer’s shear modulus and limiting stretch, respectively.

Since the dielectric elastomer is assumed to be distributed in space with a circular table surface, the deformation of the active and passive parts in the hoop direction during the actuation process is always constant, i.e., $\lambda_{a\left(c\right)}=\lambda_{a\left(p\right)}$, $\lambda_{p\left(c\right)}=\lambda_{p\left(p\right)}$. Furthermore, the radial deformation of the active and passive parts can be written as follows (see Appendix A):(5)$$\lambda_{a(r)}=\lambda_{a(p)}\sqrt{1+\frac{D^{2}}{{\left(B-A\right)}^{2}}}\;\lambda_{p(r)}=\lambda_{p(p)}\sqrt{1+\frac{{\left(L_{0}-D\right)}^{2}}{{\left(B-A\right)}^{2}}}$$
According to the geometric relationship in Figure 2d, the $sin\theta_{a}$ and $sin\theta_{p}$ can be expressed as follows:
(6)$$sin\theta_{a}=\frac{D}{\sqrt{D^{2}+{\left(B-A\right)}^{2}}}\;sin\theta_{p}=\frac{L_{0}-D}{\sqrt{{\left(L_{0}-D\right)}^{2}+{\left(B-A\right)}^{2}}}$$
The control equation that describes the actuation performance of the push-pull DEA can be obtained by combining Equations (1)–(6):(7)$$\frac{d^{2}D}{dt^{2}}+\frac{\zeta }{M}\frac{dD}{dt}+G\left(D,\Phi \right)=0$$
where the specific expression for $G\left(D,\Phi \right)$ is:
(8)$$\begin{array}{c} G\left(D,\Phi \right)=\frac{2\pi AH_{0}\mu }{M}[\left(\frac{\lambda_{a\left(r\right)}\lambda_{a\left(p\right)}^{-1}-\lambda_{a\left(r\right)}^{-3}\lambda_{a\left(p\right)}^{-3}}{1-\left(\lambda_{a\left(r\right)}^{2}+\lambda_{a\left(p\right)}^{2}+\lambda_{a\left(r\right)}^{-2}\lambda_{a\left(p\right)}^{-2}-3\right)/J_{lim}}-\frac{\epsilon \Phi^{2}}{\mu H_{0}^{2}}\lambda_{a\left(r\right)}\lambda_{a\left(p\right)}\right)sin\theta_{a}\\ -\left(\frac{\lambda_{p\left(r\right)}\lambda_{p\left(p\right)}^{-1}-\lambda_{p\left(r\right)}^{-3}\lambda_{p\left(p\right)}^{-3}}{1-\left(\lambda_{p\left(r\right)}^{2}+\lambda_{p\left(p\right)}^{2}+\lambda_{p\left(r\right)}^{-2}\lambda_{p\left(p\right)}^{-2}-3\right)/J_{lim}}\right)sin\theta_{p}] \end{array}$$

## 3. The Static Actuation Performance Analysis of Push-Pull DEA

The composition of the dielectric elastomer is one of the most important determinants of its actuation performance. The VHB 4910/4905 (3M Company, Saint Paul, MN, USA) is widely used in the development of dielectric elastomer actuators due to its superior dielectric properties. However, its significant viscoelasticity often leads to a non-negligible response delay in the actuator under dynamic excitation [40]. OPPO Band 8003^TM^ (Oppo Medical Inc., Seattle, WA, USA), a commercially available natural rubber film, is a candidate for use in dielectric elastomeric materials due to its low viscosity, good durability, and high toughness [41,42,43]. In uniaxial tensile testing, the stress–strain hysteresis area of OPPO Band 8003^TM^ was just 2.3%, whereas the VHB was up to 19.3% [41]. OPPO Band 8003^TM^ [42,43], produced by Oppo Medical Inc Company, was selected as the dielectric elastomer material for this study. It is a natural rubber material with 0.9 wt% organic filler and 0.56 wt% carbon added, and its initial thickness H_0_ is only 0.224 mm. The pre-stretch applied to the active and the passive part of the film is $\lambda_{a\left(p\right)}=1.5$ and $\lambda_{p\left(p\right)}=1.5,$ respectively. The circular mass block, with a mass of 15.16 grams and a radius of 15 mm, as well as the annular rigid frame, with an inner diameter of 35 mm, are both made from 4 mm-thick acrylic sheets that have been processed using laser cutting. The rigid support is fabricated using metal bolts with a length of L_0_ = 28 mm.

Figure 3a depicts the experimental setup of the actuator performance test. The excitation voltage is supplied by a high-voltage amplifier (Model 610E, Trek, New York, NY, USA) and a wave generator (DG4062, Rigol, Suzhou, China). The displacement response of the actuator is measured by a laser displacement sensor (LK-Navigator 2, Kenyence, Osaka, Japan) and controlled by a personal computer. The experiment discovered that OPPO Band 8003^TM^ was susceptible to electrical breakdown failure while using commercial carbon grease (MG Chemicals, Burlington, Canada). This problem was successfully prevented by utilizing a mixture of graphite and dimethyl silicone oil with a mass ratio of 3:10. Therefore, the commercial carbon grease and the custom-made electrode were deposited on the same-size film using a mask template and a brush, then kept in the same environment for 24 h. Because the modulus of the compliant electrode was much smaller than that of the dielectric elastomer, the influence of the electrode on the actuation deformation can be disregarded. However, when a solid electrode such as a hydrogel is used, the impact of electrode thickness usually needs to be considered [44]. A broken surface was discovered on the film coated with the commercial electrode, as demonstrated in Figure 3b. However, the film’s surface remained unchanged when covered with the homemade electrode. Consequently, OPPO Band 8003^TM^ is more likely to experience electrical breakdown failure when commercial electrodes are utilized. This might be because the components in Carbon Grease modify the dielectric and mechanical characteristics of the dielectric elastomer material. The displacement response of the actuator under static voltage is demonstrated in Figure 3b. The results showed that the experimental data are mostly consistent with the theoretical results, hence validating the accuracy and validity of the theoretical model. The following calculated parameters are employed in this study [42,43]: relative permittivity $\epsilon_{r}=2.7$, shear modulus μ=620 kPa, and material limit stretch properties $J_{lim}=538$.

According to the structural characteristics of the push-pull DEA, film pre-stretching $\lambda_{a\left(p\right)}$ and $\lambda_{p\left(p\right)}$, mass block radius A, rigid frame inner diameter B, and the support connector length L_0_ constitute the set of parameters for the actuator design. The critical design parameters $\lambda_{a\left(p\right)}$, $\lambda_{p\left(p\right)}$, B, and L_0_ are studied and analyzed based on the theoretical model to investigate the influence on the actuation performance and its effect under various design parameters. It is assumed that the mass block radius A = 15 mm and the actuation voltage Φ=3 kV are constant, as depicted in Figure 4.

The actuation displacement with respect to the inner diameter B of the rigid frame and length L_0_ of the support connector is shown in Figure 4a. Within the figure, the active and passive parts of the film pre-stretch are set at $\lambda_{a\left(p\right)}=1.5$ and $\lambda_{p\left(p\right)}=1.5$, respectively. The value of the actuation displacement increases with B and L_0_. For a fixed L_0_, the actuation displacement increases gradually with B. Similarly, the actuation displacement is positively related to L_0_ when B is held constant.

The effects of active part film pre-stretching $\lambda_{a\left(p\right)}$ and passive part film pre-stretching $\lambda_{p\left(p\right)}$ on the actuation displacement are illustrated in Figure 4b. The inner diameter of the rigid frame is B = 35 mm, and the length of the support frame is L_0_ = 28 mm. The figure demonstrates that the passive part film pre-stretching $\lambda_{p\left(p\right)}$ has a negligible influence on the actuator’s performance. However, the influence of the active part film pre-stretching $\lambda_{a\left(p\right)}$ is more evident because the film becomes thinner with an increase in the active part film pre-stretching. Consequently, the electric field increases at the same voltage, increasing the response.

Dielectric elastomers commonly exhibit four failure types [45] during actuation, of which loss of tension and electrical breakdown are often utilized to evaluate the maximum actuation performance of the actuator. Loss of tension refers to the nullification of single-direction stress in a dielectric elastomer film with an increase in the loading voltage, forming wrinkles. For the conical DEA structure, the hoop stress of the elastomer film gradually decreases as the excitation voltage increases, leading to wrinkling phenomena along the hoop and rapid electrical breakdown failure [46]. Therefore, the critical voltage $\Phi_{LT}$ at which the actuator undergoes loss of tension can be determined by analyzing the hoop stress $\sigma_{a\left(c\right)}$ of the active film in the push-pull DEA:(9)$$\Phi_{LT}=H_{0}\sqrt{\frac{\mu }{\epsilon }\frac{\lambda_{a\left(r\right)}^{-2}-\lambda_{a\left(p\right)}^{-4}\lambda_{a\left(r\right)}^{-4}}{1-\left(\lambda_{a\left(r\right)}^{2}+\lambda_{a\left(p\right)}^{2}+\lambda_{a\left(r\right)}^{-2}\lambda_{a\left(p\right)}^{-2}-3\right)/J_{lim}}}$$

Electrical breakdown of dielectric elastomers occurs when the electric field strength in the thickness direction of the film reaches a critical value, forming a pathway current along the thickness direction of the film and an accompanying electrical sparking phenomenon that destroys the material. Electrical breakdown in dielectric elastomers may be attributed to thermal escape due to heat accumulation [47,48]. Therefore, prolonged and continuous high-voltage loading should be avoided throughout the actuation. Experimental tests have yielded a semi-analytical expression for calculating the electric breakdown voltage of OPPO Band 8003^TM^ [42] (a similar mathematical formula has been used for the study of the electric breakdown strength of the VHB material [49,50]), as follows:(10)$$\Phi_{EB}=E_{B}H_{0}{\left(\lambda_{a\left(r\right)}\lambda_{a\left(p\right)}\right)}^{-0.52}$$
where $E_{B}$ is the electrical breakdown strength of the material when it is in a stress-free condition, and $E_{B}=97\;\mathrm{MV}/m$ is the value that is used in the study since it is based on known experimental data [42,43].

The effect of key design parameters on the actuator’s maximum deformation during actuation and critical failure voltage are shown in Figure 5. The actuator failures shown in Figure 5a,b may occur under various film pre-stretching conditions. In the figure, the tension loss failure is represented by the solid square, while the solid triangle shows the electrical breakdown failure. Figure 5c,d provides an in-depth analysis of the maximum actuation displacement and critical voltage when the actuator fails for various combinations of design parameters. As illustrated in Figure 5c, the actuator only fails due to an electrical breakdown when B = 20 mm and L_0_ = 28–34 mm, whereas all other parameter combinations result in loss of tension. Moreover, while L_0_ is constant, the bigger B is, the greater the actuation displacement created by the actuator, and the greater the critical voltage when the failure occurs. When B is constant, as L_0_ decreases, the critical failure voltage increases, while the maximum actuation displacement decreases. As observed in Figure 5d, only when the active film is pre-stretched to $\lambda_{a\left(p\right)}=1.0$, electrical breakdown is a potential failure mode for the actuator. All other parameter combinations result in failure which is attributed to the loss of tension. In addition, the higher the pre-stretched value of the active film is, the lower the critical failure voltage of the actuator, while pre-stretching of the passive film affects the maximum actuation displacement.

## 4. Dynamic Actuation and Natural Vibration Characteristics of Push-Pull DEA

When the active film in the actuator is exposed to a sinusoidal voltage $\Phi =\Phi_{ac}sin\left(2\pi ft\right)$, the governing equation describing the dynamic actuation behavior of the actuator can be rewritten as follows:
(11)$$\begin{array}{l} \frac{d^{2}D}{dt^{2}}+\frac{\zeta }{M}\frac{dD}{dt}+\frac{2\pi AH_{0}\mu }{M}[(\frac{\lambda_{a\left(r\right)}\lambda_{a\left(p\right)}^{-1}-\lambda_{a\left(r\right)}^{-3}\lambda_{a\left(p\right)}^{-3}}{1-\left(\lambda_{a\left(r\right)}^{2}+\lambda_{a\left(p\right)}^{2}+\lambda_{a\left(r\right)}^{-2}\lambda_{a\left(p\right)}^{-2}-3\right)/J_{lim}}\\ \;\;-\frac{\epsilon \Phi_{ac}^{2}{\left(sin\left(2\pi ft\right)\right)}^{2}}{\mu H_{0}^{2}}\lambda_{a\left(r\right)}\lambda_{a\left(p\right)})sin\theta_{a}\\ \;\;-(\frac{\lambda_{p\left(r\right)}\lambda_{p\left(p\right)}^{-1}-\lambda_{p\left(r\right)}^{-3}\lambda_{p\left(p\right)}^{-3}}{1-\left(\lambda_{p\left(r\right)}^{2}+\lambda_{p\left(p\right)}^{2}+\lambda_{p\left(r\right)}^{-2}\lambda_{p\left(p\right)}^{-2}-3\right)/J_{lim}})sin\theta_{p}]=0 \end{array}$$
where $\Phi_{ac}$ represents the amplitude of the voltage, and *f* denotes the excitation frequency.

Figure 6 depicts the dynamic response of the actuator with the voltage amplitude $\Phi_{ac}=2\;\mathrm{kV}$ and excitation frequency *f* = 24.1. The laser sensor measures the response displacement, and the sampling frequency is 1 kHz. Figure 6a illustrates the displacement response curve of the actuator based on experimental data and theoretical results. When the damping coefficient equals ζ=0.181 Ns/m, the theoretical analysis matches the practical test. Figure 6b displays the phase response diagram of the actuator under dynamic excitation. The phase curve begins at the initial position and forms a ring-shaped band. The limit cycle and Poincaré map are trusted methods to determine if a structure’s response is periodic, as cited in references [51,52]. To reveal more about the actuator’s response, Figure 6c analyzes the limit cycle and Poincaré map, revealing that the limit cycle is a smooth, closed curve and the Poincaré map converges to a point on the limit cycle. This confirms that the response of the actuator under dynamic stimulation is periodic.

The linear damping coefficient ζ of the dielectric elastomer actuator under dynamic driving can be expressed as a function of the excitation frequency [51]. Therefore, the response displacement of the actuator at five frequencies was experimentally measured and then fitted to obtain the linear damping coefficient ζ of the actuator as a function of the excitation frequency f (as shown in Figure 7a):(12)$$\zeta =0.1229f^{2}-5.7839f+68.1729$$

The natural vibration frequency of the structure is the fundamental parameter that is used to evaluate its dynamic response. Hence, the displacement response amplitude of the actuator was investigated via the abovementioned theoretical model at various excitation frequencies, as shown in Figure 7b. According to the results, the actuator’s displacement response amplitude peaks at the excitation frequency *f* = 23.59 Hz. This result is consistent with the experimental data. A sinusoidal excitation with a frequency range from 0.1 Hz to 50.1 Hz was used for the sweep test of the actuator to further investigate the dynamic behavior of the actuator, with a frequency change rate of 1 Hz/s and a sampling frequency of 1 kHz, as shown in Figure 7c. The displacement response curve of the actuator exhibits a peak response at approximately t = 24.02 s, and the excitation frequency at this time may be determined from the frequency rate of change to be roughly 24.02 Hz. The results of a fast Fourier transform in MATLAB applied to the time domain signal obtained from the experimental test are shown in Figure 7d. Hence, the natural frequency of the actuator can be determined more intuitively. The results demonstrate that the actuator response frequency is 47.28 Hz, i.e., approximately double the excitation frequency, which is consistent with the previous research observations [53,54]. Although a sinusoidal voltage signal is supplied to the elastomer, the actual electric field force on the elastomer is proportional to the square of the voltage signal. Consequently, a two-times relationship between the response and excitation frequency is observed.

The influence of the design parameters and the excitation signal on the dynamic properties of the actuator is analyzed in Figure 8 to investigate the resonant frequency of the actuator and the resonant response. The influence of the excitation voltage bias component $\Phi_{dc}$ on the resonance frequency and response amplitude is examined in Figure 8a. The results demonstrate that the resonant frequency of the structure is reduced while the response amplitude at the resonance increases when the bias voltage $\Phi_{dc}$ is increased. According to Figure 8b, the resonance frequency and the response amplitude of the actuators both increase with the length L_0_ of the support connection. Moreover, according to Figure 8c, the rigid frame’s inner diameter B increases, and the actuator’s resonant frequency decreases significantly. However, the displacement response amplitude at the resonant frequency shows a quadratic rule; it increases at first and then decreases as B increases. The pre-stretch $\lambda_{a\left(p\right)}$ of the active film shows a positive correlation between both the resonant frequency and the actuator’s response, as shown in Figure 8d.

## 5. Conclusions

As an extended conical dielectric elastomer actuator structure, push-pull DEA shows promise for application in controllable acoustics, haptic displays, microfluidic control, and multi-stable actuation. In this paper, an electromechanical coupling model describing the actuation performance of push-pull DEA was established based on analysis of the force of the mass block in the actuator. Moreover, the model was experimentally validated. Based on the theoretical model, the influence of key design parameters on the actuator’s actuation performance, failure type, and failure critical voltage was investigated and analyzed. The actuator’s dynamic response and natural vibration characteristics were experimentally and theoretically explored. In contrast, the effects of key design parameters and excitation signals on the resonant frequency and response amplitude were further investigated. This study enriches the design theory and techniques of dielectric elastomer actuators while providing theoretical background and practical experience for the broad application of push-pull DEA.

## Figures and Tables

**Figure 1 polymers-15-01037-f001:**
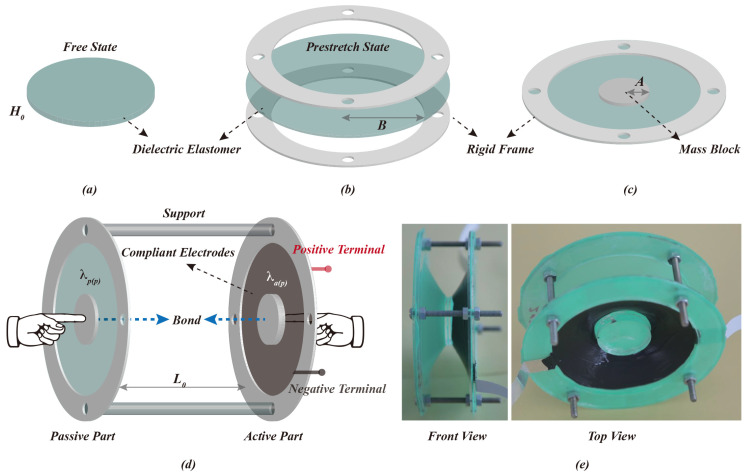
Push-pull DEA preparation process and experimental photos: (**a**–**c**): the DE film, with an initial thickness of *H_0_*, is pre-stretched and fixed to a rigid frame with an inner radius of B. A mass block with a radius of A is attached to the central position on both sides of the film; (**d**): two identical actuated units are assembled via a rigid support of length $L_{0}$. The DE film of one of the actuated units is coated with compliant electrodes, which serve as the active part, and the equal biaxial pre-stretching applied to the film is $\lambda_{a(p)}$. Meanwhile, the other actuated unit serves as the passive part, and its film is subjected to equal biaxial pre-stretching of magnitude $\lambda_{p(p)}$. The push-pull DEA is accomplished by bonding the mass blocks in the active and passive parts; (**e**): front and top views of the push-pull DEA in the experiment.

**Figure 2 polymers-15-01037-f002:**
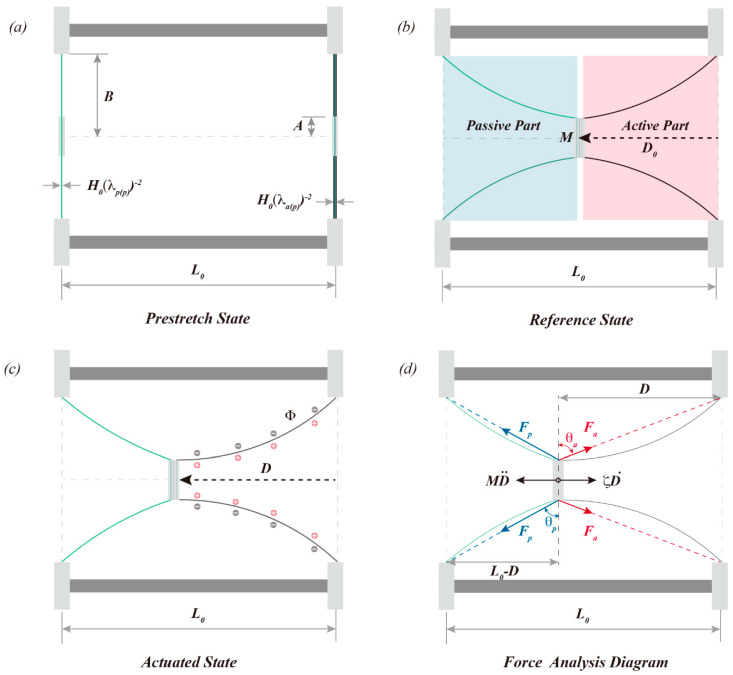
Diagram of the push-pull DEA states and mass block force analysis: (**a**–**c**) are the pre-stretch, reference, and actuation states, respectively, of the push-pull DEA; (**d**) is the force analysis diagram of the mass block in the actuator.

**Figure 3 polymers-15-01037-f003:**
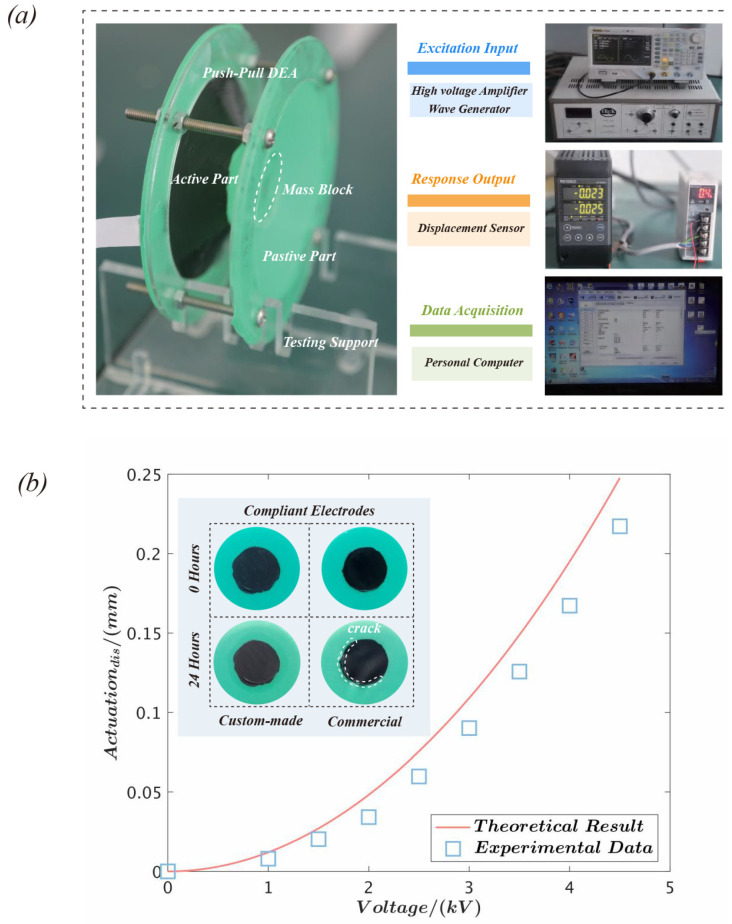
Push-pull DEA performance test and its static response: (**a**) the experimental setup for measuring the actuation performance; (**b**) the actuation displacement of the push-pull DEA at static voltage and comparison with the theoretical results.

**Figure 4 polymers-15-01037-f004:**
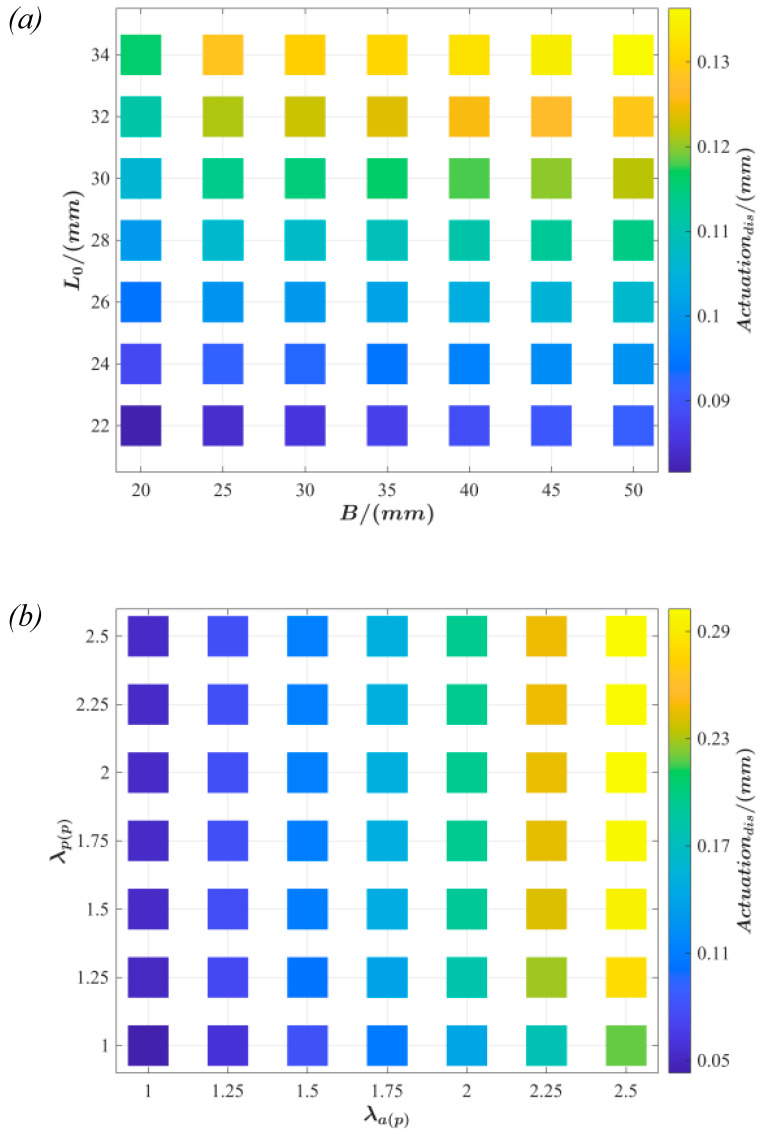
The influence of key design parameters on the actuation performance of the push-pull DEA: (**a**) the actuation performance of the push-pull DEA with the L_0_ and B parameters; (**b**) the actuation performance of the actuator by combination of $\lambda_{a\left(p\right)}$ and $\lambda_{p\left(p\right)}$ parameters.

**Figure 5 polymers-15-01037-f005:**
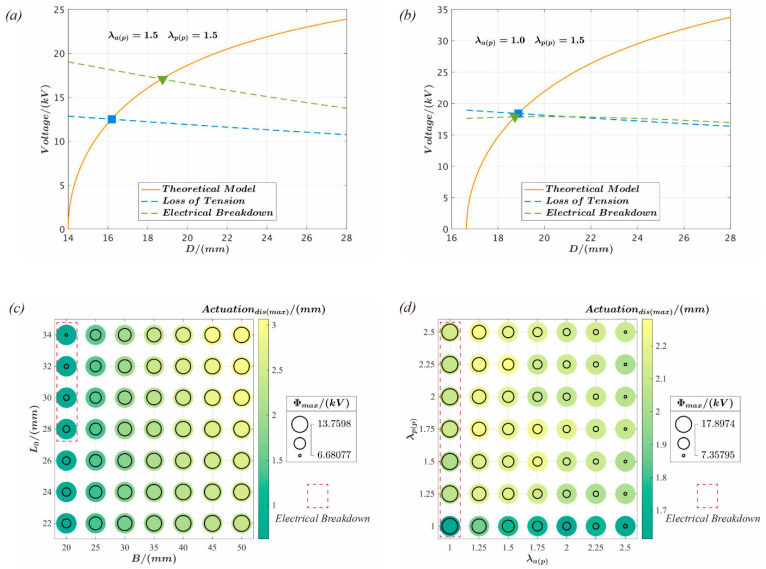
The influence law of key design parameters on the maximum actuation deformation and critical failure voltage of the actuator. (**a**,**b**) show the different types of failures that can occur with different parameter combinations, where (**a**) produces loss of tension failure and (**b**) produces electrical breakdown failure. The blue square represents a loss of tension in the actuation, while the green triangle indicates electrical breakdown that occurred during the actuation process; (**c**,**d**) provide the maximum actuation displacement and the critical voltage for actuator’s failure with different parameter combinations, respectively.

**Figure 6 polymers-15-01037-f006:**
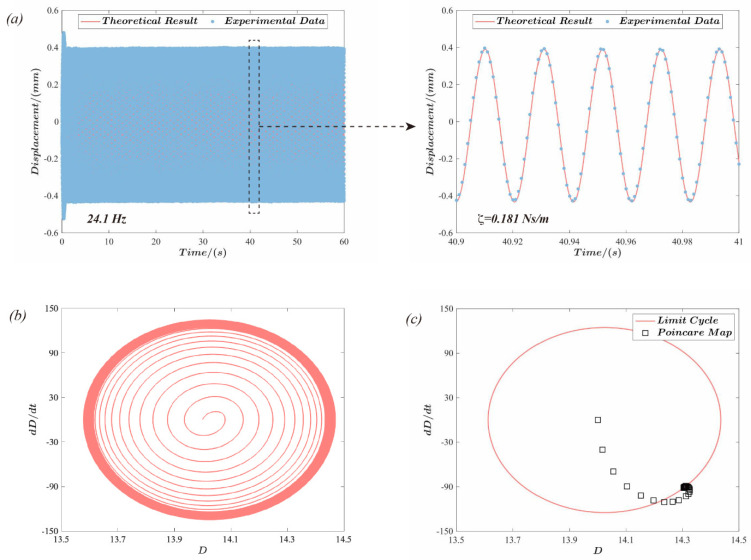
Dynamic actuation performance of the actuator under sinusoidal voltage excitation: (**a**) the response displacement versus time of the actuator for the excitation frequency *f* = 24.1 Hz and the comparison with the theoretical results; (**b**,**c**) provide the phase diagram, limit cycle, and Poincaré map of the push-pull DEA based on the theoretical analysis.

**Figure 7 polymers-15-01037-f007:**
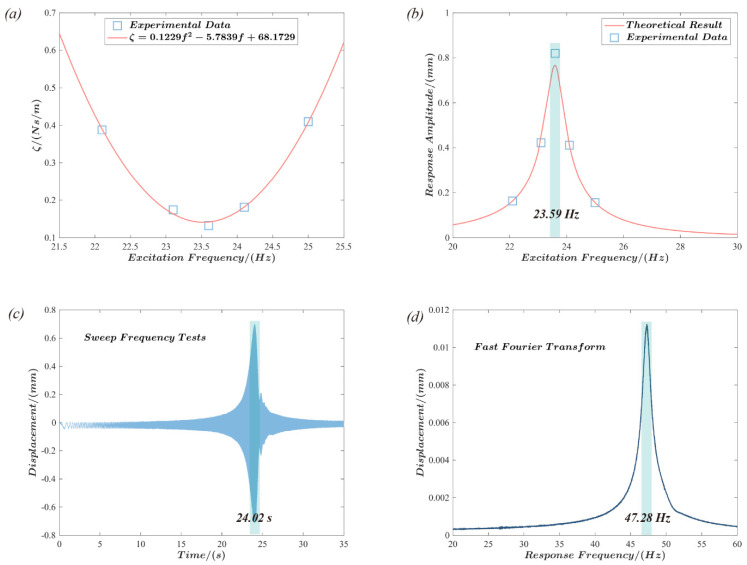
Natural vibration characteristics of the push-pull DEA. (**a**): the mathematical relationship between the linear damping coefficient ζ and the excitation frequency *f*; (**b**): theoretical analysis and experimental measurements were used to demonstrate the amplitude of the displacement response of the actuators under different excitation frequencies; the results show that the actuator exhibits a response peak at an excitation frequency of 23.59 Hz; (**c**,**d**): the time domain curves of the actuators under swept frequency tests and the frequency domain obtained after fast Fourier transformation, respectively. The time domain curve exhibits a peak at 24.02 *s*, while the frequency domain curve exhibits a peak at 47.28 Hz, which are approximately in a multiple relationship.

**Figure 8 polymers-15-01037-f008:**
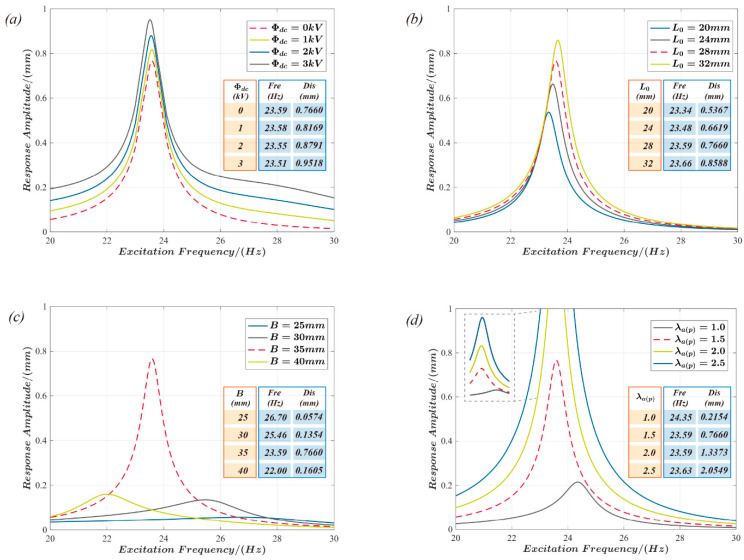
The influence of design and driving parameters on the resonant frequency and actuator’s response. (**a**–**d**): the influence of the excitation voltage bias component $\Phi_{dc}$, rigid connector length L_0_, rigid annular frame inner diameter B, and active film pre-stretch $\lambda_{a\left(p\right)}$, respectively, on the resonant frequency of the actuator and its actuation response.

## Data Availability

The data presented in this study are available on request from the corresponding author.

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
