# Peer review of "Design Analysis and Actuation Performance of a Push-Pull Dielectric Elastomer Actuator"

_polymers, 2023, doi:10.3390/polym15041037_

Round 1

Reviewer 1 Report

In the manuscript, the authors present a comparison between experimental and analytical investigations of a push-pull dielectric elastomer actuator. In general, the paper is well written and structured and shows interesting results. My main concern is, that the derivations of the analytical solutions are not presented in any form and can therefore not be easily comprehended or verified. I think that these should be provided at least in the appendix. Moreover, some of the results are discussed only very briefly which makes me question, if they are even relevant.

Detailed comments may be found in the attached pdf file.

Author Response

Dear Reviewer,

Thank you for your letter and the comments concerning our manuscript entitled “Design analysis and actuation performance of a push-pull dielectric elastomer actuator.” These comments are valuable and very helpful for revising and improving our manuscript. We have analyzed the comments and revised them carefully. We hope this revision can make our manuscript more acceptable. The revised portion is marked in red in the manuscript. The responses to the comments and the revisions are as follows:

Response to Reviewer 1 Comments

In the manuscript, the authors present a comparison between experimental and analytical investigations of a push-pull dielectric elastomer actuator. In general, the paper is well written and structured and shows interesting results. My main concern is, that the derivations of the analytical solutions are not presented in any form and can therefore not be easily comprehended or verified. I think that these should be provided at least in the appendix. Moreover, some of the results are discussed only very briefly which makes me question, if they are even relevant.

Detailed comments may be found in the attached pdf file.

Point 1: The Reviewer comments on lines 39 to 43 on the page 1 as follows: "A sketch of the geometry of a conical DEA would improve the description."

Response 1:

Thanks for kindly pointing out this comment.

The schematic diagrams in literature 23 and 24 clearly show two design methods for the conical DEA, and figure 1 in the following text also provides a clear description of the preparation process and structural characteristics of the conical DEA. Therefore, additional structural diagrams of the conical DEA will not be added in the introduction section.

Point 2: The Reviewer comments on lines 60 on the page 2 as follows: "Maybe add a paragraph on the modeling background (state of the art, etc). As far as I see it, the introduction only covers the applications."

Response 2:

Thanks for kindly pointing out this comment.

We revised the statements on page 2 of the revision and marked the changes with red font.

“Theoretical and experimental studies have been conducted to evaluate the actuation performance of conical DEAs, as reported in references [22-23, 33-35]. For instance, an electromechanical coupling model, which was developed based on thermodynamics and continua mechanics for conical DEAs, was utilized to examine the non-uniform deformation and potential failure modes of the actuator [22, 23]. The multi-mode dynamic response of conical DEAs under varying excitation frequencies was analyzed through experiments [33]. Additionally, an electromechanical-magnetic coupling model was established to study the bistable performance of conical DEAs under magnetic fields [34]. The actuation performance of conical DEAs was also analyzed using the finite element method under different excitation voltages [35]. However, research into the performance of push-pull DEAs remains limited both theoretically and experimentally.”

Point 3: The Reviewer comments on lines 63 on the page 2 as follows: "how do you apply the pre-stretch? If possible, please also add photographs."

Response 3:

Thanks for kindly pointing out this comment.

The pre-stretch of dielectric elastomer is achieved by using a biaxial thin film stretcher, which can stretch an elastic film within a certain range within a plane.

Point 4: The Reviewer comments on lines 63 on the page 2 as follows: "Also, please add information, why a biaxial pre-stretch is necessary."

Response 4:

Thanks for kindly pointing out this comment.

We revised the statements on page 2 of the revision and marked the changes with red font.

The purpose of applying a biaxial pre-stretch is to increase the electric field strength under a constant voltage by reducing the thickness of the dielectric elastomer film, thus enhancing the actuation performance of the DEA.”

Point 5: The Reviewer comments on lines 64 on the page 2 as follows: "following the sketch, "B" seems to be the inner radius."

Response 5:

Thanks for kindly pointing out this comment.

B should be the inner radius of the rigid frame, not the inner diameter.

We revised the statements on page 2 of the revision and marked the changes with red font.

A circular dielectric elastomer film with an initial thickness of H0 is pre-stretched in an equal biaxial direction and then attached to a circular rigid frame with an inner radius of B.”

Point 6: The Reviewer comments on lines 71 on the page 2 as follows: "delete "several". It is two photographs."

Response 6:

Thanks for kindly pointing out this comment.

We have deleted the word "several" at the corresponding location on page 2.

Point 7: The Reviewer comments on lines 73 on the page 2 as follows: "The caption should be reworked. Put more information into the caption so that the figure is (at best) self explanatory"

Response 7:

Thanks for kindly pointing out this comment.

We revised the statements on page 2 of the revision and marked the changes with red font.

Figure 1. Push-pull DEA preparation process and experimental photos: (a) - (c) the DE film, with an initial thickness of H0, is pre-stretched and fixed to a rigid frame with an inner radius of B. A mass block with a radius of A is attached to the central position on both sides of the film; (d) Two identical actuated parts are assembled with rigid support. The DE film of one of the actuated units is coated with flexible electrodes, serving as the active part. Meanwhile, the other actuated unit serves as the passive part. The push-pull DEA is accomplished by bonding the mass blocks in the active and passive parts; (e) Front and top views of the push-pull DEA in the experiment.”

Point 8: The Reviewer comments on lines 92 on the page 3 as follows: "A nonlinear distribution of what? Does this refer to the the displacement or the distribution of the electrical loading?"

Response 8:

Thanks for kindly pointing out this comment.

The nonlinear distribution mentioned in the text refers to the thickness of the dielectric elastomer film in the conical DEA having a nonlinear distribution.

We revised the statements on page 3 of the revision and marked the changes with red font.

“The thickness of the dielectric elastomer film in the conical DEA displays a nonlinear distribution …”

Point 9: The Reviewer comments on lines 94 on the page 3 as follows: "I am not sure, what is meant by "circular table surface". Maybe, the geometry could be described in detail or an additional sketch could help"

Response 9:

Thanks for kindly pointing out this comment.

We revised the statements on page 3 of the revision and marked the changes with red font.

“…, and the actual shape of the dielectric elastomer film in the conical DEA is similar to a hyperbolic surface, formed by the solid gray line rotating 360 degrees around the actuation direction in Figure 2 (d). However, for the convenience of analysis, it is commonly simplified as a cone surface, which is formed by the red dotted line rotating 360 degrees around the actuation direction.”

Point 10: The Reviewer comments on lines 97 on the page 4 as follows: "\zeta is not defined in the text. Also, it is written in different font in figure 2"

Response 10:

Thanks for kindly pointing out this comment.

We revised the statements on page 4 of the revision and marked the changes with red font.

“Assuming that the forces acting on the mass block by the active part and the passive part are , and , respectively, and the resistance of the mass block during the actuation process is represented by , where  represents the linear damping coefficient, the equilibrium control equation of the mass block at this moment is:”

The character in Figure 2 is indeed , but its display in Adobe Illustrator and Microsoft Office Word software is slightly different.

Point 11: The Reviewer comments on lines 104 on the page 4 as follows: " ? " (The symbol may mean the phrase is not clearly expressed).

Response 11:

Thanks for kindly pointing out this comment.

We revised the statements on page 4 of the revision and marked the changes with red font.

The force  applied on the mass block and the radial stress  in the active part and the force  applied on the mass block and the radial stress  in the passive part are related as follows [22]:”

Point 12: The Reviewer comments on the equation 3 on the page 4 as follows: "In the given reference I could not find the given equations. Please specify, where these equations are taken from exactly. "

Response 12:

Thanks for kindly pointing out this comment.

References 37 and 38 present classic theoretical studies on the electromechanical coupling behavior of dielectric elastomers. Although Equation 3 in the text cannot be found explicitly in either of these references, it can be derived from the principles of dielectric elastomer theory. For instance, Formulas 33-36 in Reference 37 and 4a-4b in Reference 38 serve as valuable resources for this derivation.

Point 13: The Reviewer comments on the equation 5 on the page 4 as follows: " How do you derive this equation? Maybe put the details in the appendix. "

Response 13:

Thanks for kindly pointing out this comment.

A detailed derivation of the equation has been added to the appendix.

Point 14: The Reviewer comments on the equation 8 on the page 5 as follows: "Details of the derivation could be put into the appendix. "

Response 14:

Thanks for kindly pointing out this comment.

A detailed derivation of the equation has been added to the appendix.

Point 15: The Reviewer comments on lines 130 on the page 5 as follows:"please correct grammar "

Response 15:

Thanks for kindly pointing out this comment.

We revised the statements on page 5 of the revision and marked the changes with red font.

The VHB 4910/4905, produced by the 3M Company, is widely used in the development of dielectric elastomer actuators due to its superior dielectric properties.”

Point 16: The Reviewer comments on lines 141 on the page 5 as follows: "The mass block is also made from OPPO Band? And what is the mass block's thickness? "

Response 16:

Thanks for kindly pointing out this comment.

The circular mass block with a mass of 15.16 grams and a radius of 15mm, as well as the annular rigid frame with an inner diameter of 35mm, are both made from 4mm thick acrylic sheets that have been processed using laser cutting.

We revised the statements on page 5 of the revision and marked the changes with red font.

The circular mass block with a mass of 15.16 grams and a radius of 15mm, as well as the annular rigid frame with an inner diameter of 35mm, are both made from 4mm thick acrylic sheets that have been processed using laser cutting.”

Point 17: The Reviewer comments on lines 152 on the page 6 as follows: "rather "custom made" "

Response 17:

Thanks for kindly pointing out this comment.

We revised the statements on page 6 of the revision and marked the changes with red font.

Therefore, the commercial carbon grease and the custom-made electrode were …”

We have also replaced "Home-made" with "Custom-made" in Figure 3(b).

Point 18: The Reviewer comments on Figure 3 on the page 6 as follows: "Maybe, the actuation could be given in terms of a stretch of the material. In terms of the displacement, I can not understand, of 0.2 mm are a large deformation or not."

Response 18:

Thanks for kindly pointing out this comment.

The actuation displacement of the DEA is closely linked to several factors, such as the material of the elastomer, the geometry of the actuator, the pre-stretch of the film, and the applied voltage. In Figure 3, natural rubber was utilized as the dielectric elastomer material, and the initial pre-stretch of the film was 1.5, which resulted in a displacement of only 0.22mm under a voltage of 4.5kV. The primary aim of this experiment was to validate the accuracy and reliability of the theoretical model, providing a solid theoretical foundation for further research. By increasing either the applied voltage or the pre-stretch, the displacement response of the actuator can be significantly improved.

Point 19: The Reviewer comments on lines 186 on the page 7 as follows: "delete the word  "actuation" "

Response 19:

Thanks for kindly pointing out this comment.

We have deleted the word "actuation" at the corresponding location on page 7.

Point 20: The Reviewer comments on Figure 4(a) on the page 8 as follows: "please give the displacement here in mm as well, so it is consistent with Figure 3 "

Response 20:

Thanks for kindly pointing out this comment.

We have already modified Figure 4 in the revised version.

Point 21: The Reviewer comments on Figure 4(a) on the page 8 as follows: "If I understand correctly, the values in the experiment are L0=28 mm and B=35 mm. At a voltage difference of 3 kV the experiment and the calculation in Figure 3 result in a displacement <0.1 mm. Why is the displacement here >0.1 mm?"

Response 21:

Thanks for kindly pointing out this comment.

In Figure 3(b), the experimental values of L0 and B are 28mm and 35mm, respectively, and the actuation displacement is 0.0901mm. The theoretical results in Figure 3(b) have a displacement of 0.1090mm under a voltage of 3kV. Figure 4(a) is the result of the theoretical analysis, so it can be seen from Figure 4(a) that under the parameter combination of B=35mm and L0=28mm, the actuation displacement is more significant than 0.1mm.

Point 22: The Reviewer comments on the equation 9 on the page 9 as follows: "Again, please show the derivation of this equaiton in more detail (maybe in the appendix) "

Response 22:

Thanks for kindly pointing out this comment.

A detailed derivation of the equation has been added to the appendix.

Point 23: The Reviewer comments on lines 249 to 252 on the page 11 as follows: "This is a very brief discussion of the results. Please provide a more detailed analysis of the plots."

Response 23:

Thanks for kindly pointing out this comment.

We revised the statements on page 11 of the revision and marked the changes with red font.

“Figure 6(b) displays the phase response diagram of the actuator under dynamic excitation. The phase curve begins at the initial position and forms a ring-shaped band. The limit cycle and Poincaré map are trusted methods to determine if a structure's response is periodic, as cited in references [51,52]. To reveal more about the actuator's response, Figure 6(c) analyzes the limit cycle and Poincaré map, revealing that the limit cycle is a smooth closed curve and the Poincaré map converges to a point on the limit cycle. This confirms that the response of the actuator under dynamic stimulation is periodic."

Point 24: The Reviewer comments on Figure 6(a) on the page 8 as follows: "This part of the figure seems senseless. Could you explain, why the left plot is shown?"

Response 24:

Thanks for kindly pointing out this comment.

The left graph in Figure 6(a) is intended to compare the difference between the experimental data and theoretical result throughout the testing period, while the right graph displays the difference between the two by selecting a small time span.

Point 25: The Reviewer comments on lines 273 to 274 on the page 12 as follows: "That is an odd coincidence. Could you please double check if the frequency is correct."

Response 25:

Thanks for kindly pointing out this comment.

Due to the frequency change rate of 1Hz/s in the sweep experiment, the excitation frequency corresponding to the peak response of the actuator at 24.02 seconds is approximately 24.02Hz. Through subsequent fast Fourier transforms, the response frequency of the actuator was found to be 47.28Hz, with the response frequency of the actuator approximately twice the excitation frequency. The same conclusion has been reported in previous studies [53, 54].

[53] Fox J, Goulbourne N. On the dynamic electromechanical loading of dielectric elastomer membranes. J Mech Phys Solids 2008;56:2669–86. https://doi.org/10.1016/j.jmps.2008.03.007.

[54] Zhu J, Cai S, Suo Z. Nonlinear oscillation of a dielectric elastomer balloon. Polym Int 2010;59:378–83. https://doi.org/10.1002/pi.2767.

Thank you again for your tremendous support and friendship.

Best Wishes.

Reviewer 2 Report

This paper describes actuation performance of a conical push-pull dielectric elastomer actuator (DEA) by experimentally and theoretically. DEA has been extensively investigated due to their excellent actuation properties. It is not recommended to publish in the Polymers journal in its present form unless the author reveals at least the following:

1) On page 2, in Figure 1, the length of the line of B is miss match with the inner diameter, it is a radius of inner circle.

2) On page 6, at line 1 (152), it is important that the thickness of electrode which was home-made graphite/dimethyl silicone in this experiment. It is worth to describe the thickness of electrode or the process of coating. Is there no effect for the mechanical property by the electrode?

Author Response

Dear Reviewer,

Thank you for your letter and the comments concerning our manuscript entitled “Design analysis and actuation performance of a push-pull dielectric elastomer actuator.” These comments are valuable and very helpful for revising and improving our manuscript. We have analyzed the comments and revised them carefully. We hope this revision can make our manuscript more acceptable. The revised portion is marked in red in the manuscript. The responses to the comments and the revisions are as follows:

Response to Reviewer 2 Comments

This paper describes actuation performance of a conical push-pull dielectric elastomer actuator (DEA) by experimentally and theoretically. DEA has been extensively investigated due to their excellent actuation properties. It is not recommended to publish in the Polymers journal in its present form unless the author reveals at least the following:

Point 1: On page 2, in Figure 1, the length of the line of B is miss match with the inner diameter, it is a radius of inner circle.

Response 1:

Thanks for kindly pointing out this comment.

B should be the inner radius of the rigid frame, not the inner diameter.

We revised the statements on page 2 of the revision and marked the changes with red font.

A circular dielectric elastomer film with an initial thickness of H0 is pre-stretched in an equal biaxial direction and then attached to a circular rigid frame with an inner radius of B.”

Point 2: On page 6, at line 1 (152), it is important that the thickness of electrode which was home-made graphite/dimethyl silicone in this experiment. It is worth to describe the thickness of electrode or the process of coating. Is there no effect for the mechanical property by the electrode?

Response 2:

Thanks for kindly pointing out this comment.

The coating process for the compliant electrode was done using a mask template and a brush. Compared to the dielectric elastomer, the impact of the compliant electrode on the driving deformation of the dielectric elastomer is typically negligible, but when using a solid electrode, such as hydrogel, the effect of electrode thickness often needs to be considered.

We revised the statements on page 6 of the revision and marked the changes with red font.

“Therefore, the commercial carbon grease and the homemade electrode were deposited on the same-size film using a mask template and a brush, then kept it in the same environment for 24 hours. Because the modulus of the compliant electrode is much smaller than that of the dielectric elastomer, the influence of the electrode on the actuation deformation can be disregarded. However, when a solid electrode such as a hydrogel is used, the impact of electrode thickness usually needs to be considered [42].”

Thank you again for your tremendous support and friendship.

Best Wishes.
